# The Antitumour Mechanisms of Carotenoids: A Comprehensive Review

**DOI:** 10.3390/antiox13091060

**Published:** 2024-08-30

**Authors:** Andrés Baeza-Morales, Miguel Medina-García, Pascual Martínez-Peinado, Sandra Pascual-García, Carolina Pujalte-Satorre, Ana Belén López-Jaén, Rosa María Martínez-Espinosa, José Miguel Sempere-Ortells

**Affiliations:** 1Immunology, Cellular and Developmental Biology Group, Department of Biotechnology, University of Alicante, Ap. 99, E-03080 Alicante, Spain; andres.baeza@ua.es (A.B.-M.); miguel.medina@ua.es (M.M.-G.); pascual.martinez@ua.es (P.M.-P.); sandra.pascual@ua.es (S.P.-G.); cps58@alu.ua.es (C.P.-S.); ana.belen.lopez@ua.es (A.B.L.-J.); 2Biochemistry and Molecular Biology and Edaphology and Agricultural Chemistry Department, Faculty of Sciences, University of Alicante, Ap. 99, E-03080 Alicante, Spain; rosa.martinez@ua.es; 3Applied Biochemistry Research Group, Multidisciplinary Institute for Environmental Studies “Ramón Margalef”, University of Alicante, Ap. 99, E-03080 Alicante, Spain

**Keywords:** antioxidant activity, antitumour, apoptosis, carotenoid, cell proliferation, metastasis inhibition, oxidative stress

## Abstract

Carotenoids, known for their antioxidant properties, have garnered significant attention for their potential antitumour activities. This comprehensive review aims to elucidate the diverse mechanisms by which carotenoids exert antitumour effects, focusing on both well-established and novel findings. We explore their role in inducing apoptosis, inhibiting cell cycle progression and preventing metastasis by affecting oncogenic and tumour suppressor proteins. The review also explores the pro-oxidant function of carotenoids within cancer cells. In fact, although their overall contribution to cellular antioxidant defences is well known and significant, some carotenoids can exhibit pro-oxidant effects under certain conditions and are able to elevate reactive oxygen species (ROS) levels in tumoural cells, triggering mitochondrial pathways that would lead to cell death. The final balance between their antioxidant and pro-oxidant activities depends on several factors, including the specific carotenoid, its concentration and the redox environment of the cell. Clinical trials are discussed, highlighting the conflicting results of carotenoids in cancer treatment and the importance of personalized approaches. Emerging research on rare carotenoids like bacterioruberin showcases their superior antioxidant capacity and selective cytotoxicity against aggressive cancer subtypes, such as triple-negative breast cancer. Future directions include innovative delivery systems, novel combinations and personalized treatments, aiming to enhance the therapeutic potential of carotenoids. This review highlights the promising yet complex landscape of carotenoid-based cancer therapies, calling for continued research and clinical exploration.

## 1. Introduction

Cancer is a complex and multifactorial disease that arises from genetic and epigenetic alterations in normal cells, leading to uncontrolled growth and proliferation [1]. It is one of the leading causes of death worldwide, accounting for over 9 million deaths annually, and its incidence is projected to increase in the coming decades due to ageing populations and lifestyle changes [2].

Cancer development and progression involve complex genetic, epigenetic and environmental changes. Oxidative stress, a hallmark of cancer, results from an imbalance between reactive oxygen species (ROS) production and antioxidant defence [1,3]. ROS can damage DNA, proteins and lipids, leading to mutations, inflammation and cell death, contributing to cancer initiation and progression. Antioxidants have been proposed in studies to potentially prevent and treat cancer by mitigating ROS-induced damage [4,5,6].

Antioxidants are molecules that can scavenge ROS and neutralize their harmful effects, thereby protecting cells from oxidative damage. They include enzymes such as superoxide dismutase (SOD), catalase and glutathione peroxidase, as well as non-enzymatic compounds such as vitamins C and E, flavonoids and carotenoids [3]. In this context, one approach to preventing or treating cancer is using natural compounds with antioxidant properties, such as carotenoids [4].

Carotenoids are a group of natural pigments synthesized by plants, algae, fungi and some bacteria and archaea that have received increasing attention for their potential anticancer properties [5]. Numerous beneficial health effects of carotenoids have been described in humans, including immune system modulation, anti-inflammatory properties, eye health and cardiovascular disease prevention [6,7].

In addition, the anticancer properties of carotenoids have been extensively studied in both preclinical and clinical studies. As antioxidant molecules, carotenoids can scavenge ROS and reactive nitrogen species (RNS), preventing oxidative damage to DNA, lipids and proteins [8,9]. They can also modulate gene expression and signalling pathways involved in cell growth and differentiation, inhibit inflammation and enhance the immune response [10]. Moreover, some carotenoids have been shown to induce apoptosis, inhibit angiogenesis and block cell cycle progression in cancer cells [7,11]. The most abundant carotenoids in nature are C_40_, such as β-carotene and lutein, and C_20_, like crocetin. However, there are also less abundant carotenoids with more carbon atoms, such as bacterioruberin, a C_50_ carotenoid [12]. Among the different C_20_ and C_40_ carotenoids, lutein, fucoxanthin, β-carotene, lycopene, astaxanthin, capsanthin and crocetin have been particularly well-studied for their potential role in cancer prevention and treatment [11,13,14].

Several epidemiological studies have investigated the relationship between carotenoid intake and cancer risk, with mixed results. While some studies have reported an inverse association between carotenoid consumption and cancer incidence, others have found no significant effect or even a harmful one due to high-dose carotenoid supplements [15,16].

Overall, the potential of carotenoids as natural antioxidant weapons against cancer is an area of ongoing research. In this review, we will discuss the current evidence for the therapeutic applications of carotenoids in cancer, focusing on the mechanisms underlying their anticancer effects and the potential for future research in this field.

## 2. Biochemistry of the Carotenoids

### 2.1. Structure and Diversity

Carotenoids are terpenoid compounds composed of isoprene units, showing a variable number of conjugated doble bounds, which are responsible for their characteristic coloration [17]. Most of the carotenoids are composed of a central carbon chain of alternating single and double bonds and carry various cyclic or acyclic end groups, which makes it a remarkably diverse group structurally [18]. Variations in the number and position of double bonds and the presence of functional groups give rise to different carotenoid subclasses, including carotenes (hydrocarbons) and xanthophylls (oxygenated derivatives) [19]. For example, β-carotene consists of two β-ionone rings connected by a polyene chain with 11 conjugated double bonds, while lutein, a xanthophyll, contains an oxygenated cyclic end group (C_40_) [18] (Figure 1). There is also another type of carotenoids called apocarotenoids, which are derived from carotenes or xanthophylls by oxidative cleavage, catalyzed by carotenoid cleavage dioxygenases (CCDs) [19].

The diverse structures of carotenoids influence their physicochemical properties, such as solubility, stability and susceptibility to oxidation, as well as their biological functions [20,21,22]. The presence and position of specific functional groups within carotenoids can affect their biological activities, including antioxidant capacity, signalling and light-harvesting properties [23]. Carotenoids exhibit a range of colours, from yellow and orange to red and purple [22]. Their conjugated double bonds allow them to absorb light and act as accessory pigments in photosynthetic organisms, participating in light energy transfer processes [19].

### 2.2. Antioxidant and Pro-Oxidant Function

The antioxidant activity of carotenoids protects cells from oxidative stress induced by ROS [18]. As potent scavengers of singlet oxygen and other free radicals, carotenoids can effectively neutralize these harmful species, thereby preventing lipid peroxidation and oxidative damage to cellular components. Carotenoids (CARs) can scavenge radicals thanks to three different reactions [24]:Electron transfer (oxidation/reduction): CAR + R· → CAR·+ + R-;Hydrogen abstraction: CAR + R· → CAR· + RH;Addition: CAR + R· → R-CAR·.

The antioxidant capacity of carotenoids is influenced by their chemical structure, particularly the presence and position of conjugated double bonds. Carotenoids with a higher number of conjugated double bonds, such as β-carotene and lycopene, exhibit greater antioxidant activity due to their ability to delocalize and stabilize free radicals [20,21]. In the case of C_50_ carotenoids, they can have up to 13 conjugated double bonds (e.g., bacterioruberin), which endows them with a much higher antioxidant capacity than common C_40_ carotenoids [25,26]. Additionally, the presence of functional groups, such as hydroxyl or keto groups, can enhance the antioxidant properties of certain carotenoids [23].

However, under certain conditions, carotenoids can exhibit pro-oxidant properties, particularly in the presence of transition metal ions or in an oxygen-rich environment [27]. The pro-oxidant effects of carotenoids are attributed to their ability to undergo oxidation, leading to the generation of reactive species that can cause oxidative damage to biomolecules [28,29].

It is worth noting that while carotenoids can exhibit pro-oxidant effects under certain conditions, their overall contribution to cellular antioxidant defences are significant. The balance between their antioxidant and pro-oxidant activities depends on several factors, including the specific carotenoid, its concentration and the redox environment of the cell [30].

### 2.3. Bioavailability and Biotransformation

Understanding the bioavailability and biotransformation of carotenoids is crucial for assessing their nutritional and health benefits. The bioavailability of carotenoids is a complex process influenced by various factors, including the food matrix, processing methods and individual variations [31]. Carotenoids present in food are typically bound to the cellular matrix and require efficient digestion, emulsification and micellar solubilization for absorption in the small intestine [32]. In general terms, their bioavailability is generally considered to be low due to several factors, including their absorption efficiency, metabolism and the presence of other dietary components that may affect their uptake [33]. 

During the digestive process, carotenoids are released from the food matrix through the action of digestive enzymes, such as pancreatic lipase and bile salts, which facilitate their emulsification [34]. The liberated carotenoids then form micelles in dietary fats and bile salts, allowing for their solubilization and subsequent uptake by enterocytes [35]. The efficiency of these processes can vary depending on factors such as the chemical composition, the physical form of carotenoids (e.g., crystalline, or amorphous), the presence of other dietary components and individual factors [35]. The solubility of carotenoids also plays a critical role in their absorption and effectiveness within biological systems. Carotenoids are predominantly fat-soluble, which means they dissolve in lipids and fats rather than in water. This characteristic affects their bioavailability, as they are best absorbed when consumed with dietary fats [33].

Once absorbed, carotenoids are incorporated into chylomicrons, which are large lipoprotein particles and transported via the lymphatic system [35]. This lymphatic transport bypasses the liver, allowing carotenoids to be directly delivered to peripheral tissues. The efficiency of carotenoid incorporation into chylomicrons can be influenced by factors such as the length and saturation of dietary fatty acids, which can affect their lipophilicity and subsequent transport [36]. Furthermore, the liver metabolizes a portion of these carotenoids before they reach systemic circulation, reducing their overall bioavailability [37]. Thus, first-pass metabolism significantly impacts carotenoids’ concentration in plasma.

Within tissues, carotenoids may undergo biotransformation, including enzymatic oxidation, cleavage and conjugation reactions [24]. These transformations can generate bioactive metabolites with distinct biological activities [19]. For example, β-carotene can be enzymatically cleaved by β-carotene 15,15′-dioxygenase to yield retinol (vitamin A), which is involved in visual processes and cellular differentiation [38]. Additionally, oxidative cleavage of carotenoids can generate apocarotenoids, such as retinoic acid, which plays a crucial role in various physiological processes, including embryonic development and cellular differentiation [19]. 

## 3. Oxidative Stress and Cancer

Oxidative stress, characterized by an imbalance between the production of ROS and the ability of antioxidant defences to neutralize them, has emerged as a critical factor in cancer development and progression [39,40]. ROS are highly reactive molecules that include superoxide anion, hydrogen peroxide and hydroxyl radicals, and they are generated during normal cellular metabolism as well as in response to various environmental factors, such as radiation and chemical exposure [41]. While low levels of ROS play essential roles in cell signalling and physiological processes, excessive ROS production can inflict severe damage to cellular components, including DNA, proteins and lipids [42].

Cancer cells frequently exhibit elevated levels of ROS, a consequence of increased metabolic activity, mitochondrial dysfunction and oncogenic signalling. Enhanced metabolic processes, whether through heightened glycolysis or oxidative phosphorylation, lead to greater ROS production as a by-product [43,44]. Additionally, dysfunctional mitochondria in cancer cells contribute to excessive ROS generation, often driven by mutations or altered electron transport chain activity [45]. Oncogenes like c-Myc and RAS further exacerbate ROS levels by up-regulating pathways that generate ROS or affecting mitochondrial function [44]. The tumour microenvironment, characterized by hypoxia, inflammation and nutrient deprivation, also influences ROS levels, often increasing them [46]. Despite these elevated ROS levels, cancer cells typically develop robust antioxidant defences to mitigate oxidative damage while maintaining ROS at levels that support survival, proliferation and resistance to apoptosis [45].

ROS-induced oxidative damage can lead to the formation of DNA adducts, single-strand breaks and double-strand breaks, all of which contribute to genomic instability [47,48]. This characteristic is a hallmark of cancer and plays a critical role in driving tumour heterogeneity and promoting the acquisition of oncogenic mutations [1]. Additionally, ROS-induced DNA damage can activate DNA repair pathways, which, if not properly regulated, may increase the likelihood of survival and proliferation of genetically altered cancer cells, leading to tumour progression [42].

Moreover, ROS can act as signalling molecules, modulating various cellular processes and pathways that facilitate tumour growth and survival. For instance, ROS can activate key signalling pathways such as nuclear factor-kappa B (NF-κB), mitogen-activated protein kinase (MAPK) and phosphoinositide 3-kinase/protein kinase B (PI3K/Akt), all of which are associated with cell survival, proliferation and migration [49,50,51]. By promoting the activation of these pathways, ROS contribute to the evasion of apoptosis, enhancement of cell cycle progression and acquisition of a migratory phenotype, all of which are essential for tumour cells to survive and spread [15].

ROS play a pivotal role in shaping the tumour microenvironment, comprising components like cancer-associated fibroblasts, immune cells, endothelial cells and the extracellular matrix [52]. They promote angiogenesis, providing oxygen and nutrients for tumour growth, and activate proinflammatory pathways, leading to chronic inflammation within the tumour microenvironment [47]. Chronic inflammation promotes tumour progression by stimulating cell proliferation, angiogenesis and immune evasion [47]. Moreover, ROS presence can impair the function of immune cells like cytotoxic T cells and natural killer cells, contributing to immune evasion by the tumour and facilitating unchecked tumour proliferation [47].

All this, coupled with the fact that tumour cells can tolerate higher concentrations of ROS than healthy cells, provides an advantage for the promotion of cancer cell growth, metastasis and angiogenesis [40]. However, when ROS production reaches a level that cancer cells cannot handle, it kills them. Among the toxic effects that can be triggered, in addition to DNA damage, ROS can cause lipid peroxidation which generates cytotoxic aldehydes, such as levuglandins [53]. Additionally, ROS can damage protein structures, causing them to undergo conformational changes or lose their function [54].

In this context, the cancer cell environment, characterized by elevated ROS levels, could potentially support the pro-oxidant action of carotenoids. These compounds have the potential to increase ROS production further, pushing cancer cells beyond their tolerance threshold and inducing cell death [55]. This pro-oxidant effect could be advantageous in targeting cancer cells specifically, as they are already set to be more susceptible to ROS-induced damage due to their higher baseline ROS levels [56]. On the other hand, in healthy tissues, carotenoids act primarily as antioxidants, mitigating the side effects of cancer treatment by scavenging free radicals and preserving cellular integrity [14] (Figure 2).

However, the interplay between the antioxidant and pro-oxidant functions of carotenoids remains intricate and multifaceted. The conditions under which carotenoids act as antioxidants or pro-oxidants are contingent on factors such as the concentration, the cellular microenvironment and the type of carotenoid present [55]. At high concentrations and under specific conditions, such as unbalanced and elevated intracellular oxidative stress (often found in cancer cells), high oxygen levels (as in the lungs of smokers), reduced levels of endogenous enzymes (e.g., SOD) and increased concentrations of reactive metal ions (e.g., Fe(III) and Cu(II)), carotenoids can act as pro-oxidants [55]. Furthermore, the specific pathways driving the transition from pro-oxidant effects to the activation of apoptotic responses require further elucidation. Accordingly, ongoing research is directed toward comprehensively unravelling the underlying mechanisms regulating the duality of carotenoids actions in oxidative stress modulation within the complex landscape of cancer biology [55].

## 4. Direct Antitumoural Effects of Carotenoids on Cancer Cells

Carotenoids, as natural antioxidants, exhibit a wide range of direct antitumoural effects on cancer cells, contributing to their potential as promising therapeutic agents in cancer treatment (Figure 3). Although over 700 carotenoids have been characterized, this review focuses on the most extensively studied carotenoids in cancer research: β-carotene, lycopene, lutein, astaxanthin, capsanthin, fucoxanthin and crocin/crocetin. Our aim is to explore the mechanisms by which these carotenoids exert their anticancer effects directly on tumour cells.

### 4.1. Cell Cycle Progression and Antiproliferation

The direct impact of carotenoids on cancer cells extends to their ability to regulate cell cycle progression, presenting a potent avenue for antiproliferative interventions. The following carotenoids, β-carotene, lycopene, lutein, astaxanthin, capsanthin, fucoxanthin and crocin/crocetin, have demonstrated significant influence on cell cycle dynamics, contributing to their potential in curbing uncontrolled growth. The orchestrated influence of these diverse carotenoids on various cell cycle phases underscores their potential as formidable tools against unregulated cell growth. By intervening at distinct points in the cycle, they establish comprehensive control over cancer cell proliferation, marking their importance in therapeutic strategies aimed at reducing tumour expansion. A summary of preclinical studies on the antiproliferative effect of the carotenoids most investigated in cancer is presented in Appendix A.

#### 4.1.1. Role of β-Carotene in Cell Cycle Regulation and Antiproliferative Effects on Cancer Cells

β-Carotene is one of the most abundant carotenoids in nature, mainly found in fungi, plants and fruits. It serves as a precursor to vitamin A and plays a pivotal role in regulating the cell cycle by engaging with critical regulatory proteins. It has demonstrated its ability to hinder cell proliferation in certain cancer cell lines. For instance, in triple-negative human breast cancer cells (MDMA-MB-231), β-carotene disrupts the cell cycle involving c-jun-N-terminal kinase (JNK) intracellular pathway, leading to an arrest in the S phase [57]. The JNK pathway, belonging to the MAPK subfamily, exerts a multifaceted influence in critical aspects of cancer development, including apoptosis, cell survival, proliferation, invasion and migration [58]. 

Furthermore, β-carotene exhibits inhibitory effects on the proliferation and viability of myeloid leukemia cells (K562) in a dose- and time-dependent manner [59]. This inhibition results in a cell cycle arrest in the G0/G1 phase. Importantly, this restraint of leukemia cell growth is associated with the involvement of PPARγ (peroxisome proliferator-activated receptor gamma) and Keap1-Nrf2/EpRE/ARE (kelch-like ECH-associated protein 1-nuclear factor erythroid 2-related factor 2/electrophile responsive element/antioxidant responsive element) signalling pathways. PPARγ pathways are recognized for their role in suppressing cancer cell proliferation and growth [60], while Nrf2 functions as a significant transcription factor within the Keap1-Nrf2/EpRE/ARE pathway, contributing to the maintenance of cellular redox homeostasis and providing protection against damage and tumourigenesis [61].

Regarding the impact of β-carotene on cell cycle regulation, it can increase expression of p21. P21 is an inhibitor of cyclin-dependent kinases (CDKs) known to induce cell cycle arrest at the G0/G1 phase [62]. This mechanism has been observed in human promyelocytic leukemia HL-60 cells treated with β-carotene [63]. The treatment induces a notable increase in ROS production, accompanied by elevated p21 expression and a subsequent cell cycle arrest in the G0/G1 phase. A similar effect has been described in various human colon adenocarcinoma cell types (COLO 320 HSR, LS-174, HT-29 and WiDr) [64]. β-Carotene impedes the propagation of these cell lines by inducing cell cycle arrest in G2/M phase with a reduction in the cyclin A levels, a key regulator of the cell cycle process [65]. In xenograft mice injected with HCT-116 colon cancer cells, β-carotene supplementation suppressed tumour volume and delayed tumour formation [66].

On ACTH-secreting pituitary adenoma cells (AtT20 cells), β-carotene exerts inhibitory effects on cell proliferation and colony formation [67]. In this case, β-carotene induces cell cycle arrest, particularly in the G2/M phase, by decreasing the expression of the F-box protein S-phase kinase-associated protein 2 (Skp2) and increasing the expression of p27^Kip1^. Skp2 acts as an oncogene by targeting p27 for degradation. This molecule is overexpressed in many different human cancers, playing an important role in cancer progression; therefore, it could be a potential molecular target [68]. On the other hand, p27 binds to and restrains cyclin-CDK to stop cell cycle progression [69]. Additionally, p27 regulates various processes such as cell migration and development, independently of its CDK-inhibitory function.

Additionally, β-carotene treatment induces antiproliferative effects on SPC212 malignant mesothelioma cells [70]. These findings collectively emphasize the multifaceted role of β-carotene in regulating the cell cycle and its potential as an agent in cancer intervention and prevention.

#### 4.1.2. Role of Lycopene in Cell Cycle Regulation and Antiproliferative Effects on Cancer Cells

Abundant in tomatoes, lycopene exerts its antiproliferative influence by modulating cell cycle progression. Through arresting the cell cycle, particularly in the G0/G1 phase, lycopene impedes the advance of cancer cells and contributes to tumour growth inhibition. 

In the case of breast cancer, lycopene has been found to exert its antiproliferative influence on cancer cells by modulating cell cycle progression. This molecule increases the percentage of cells in the G0/G1 phase in the human breast adenocarcinoma cell lines MCF-7 and MDA-MB-235 [71]. Moreover, lycopene cause cell growth inhibition in MDA-MB-231 (ER/HER2-negative) and BT474 (ER-negative/HER2-positive) cells and induce G0/G1 phase arrest in both by the down-regulation of Skp2 [72], which acts as an oncogene through targeting p27 for degradation as mentioned before [68,69].

Lycopene’s ability to inhibit the proliferation of mammary cancer cells, such as the MCF-7 cell line, extends to its interference with cell cycle progression and the inactivation of the signalling pathway of insulin-like growth factors I (IGF-I), a family of proteins that hold a pivotal role in regulating cellular growth, proliferation and survival [73]. However, when this intricate IGF signalling pathway experiences dysregulation due to factors such as overproduction, genetic mutations, or crosstalk with other cellular pathways, it can significantly contribute to the development of cancer and the emergence of treatment resistance [74].

Within the same research context, it has also been demonstrated that lycopene possesses the capability to induce cell cycle arrest at the G1 phase, effectively stopping DNA synthesis in human breast cancer cells, including MCF-7 and T-47D, and in endometrial cancer cells ECC-1 [75]. This arrest of the cell cycle was orchestrated through a reduction in the abundance of cyclin D1 and D3 proteins. It is worth noting that cyclin D1 is a well-known oncogene [76] and a pivotal component in the progression of the cell cycle, with its overexpression frequently observed in various cancer cell lines and tumours, particularly in breast cancer [77,78].

Another pathway involved in the antiproliferative properties of lycopene is the NF-κB signalling pathway. Lycopene significantly can inhibit prostate (PC3 cell line) and breast cancer (MDA-MB-231 cell line) cell growth through the inhibition of IκB kinase to suppress the NF-κB signalling pathway [79]. NF-κB orchestrates the transcriptional activation of key genes involved in regulating cell proliferation, apoptosis, metastasis, angiogenesis, inflammation, cellular adhesion and invasion [80].

One study compared the antiproliferative effect of lycopene on cell lines of different cancer types (breast, colon and prostate) [81]. In the case of colorectal adenocarcinoma (HT-29), colon carcinoma (T84) and luminal epithelial breast cells (MCF-7), a notable reduction in viable cells was observed following a 48 h lycopene treatment, accompanied by a cell cycle arrest specifically at the G1 phase. However, in the context of prostate cancer represented by the DU145 cell line, the observed cell cycle arrest occurred at the M/G2 phase. These findings collectively suggest that lycopene may exert varying influences on cell cycle regulatory proteins, contingent upon the specific cancer type under consideration. 

In a mouse xenograft model, the consumption of lycopene inhibits the growth and progression of colon cancer cells [82]. The inhibitory effects of lycopene are associated with the augment of p21 protein.

Lycopene has exhibited remarkable antiproliferative capabilities in the context of leukemia cells. Treatment reduces growth of HL-60 cells in a concentration-dependent manner [83]. Furthermore, it was observed that lycopene treatment led to a pronounced inhibition of cell cycle progression specifically in the G0/G1 phase.

In the context of gastric cancer, lycopene has been shown to be a promising agent for slowing the growth of cancer cells while avoiding damage to normal gastric epithelial cells. Lycopene suppresses the growth of gastric cancer cells (AGS and SGC-7901 cell lines) while leaving normal gastric epithelial cells (GES-1 cell line) undamaged [84]. Specifically, the cell cycle of SGC-7901 cells is arrested in the G2/M phase, coinciding with an increase in the levels of CDKN1B and CDKN2B, key regulators of the cell cycle. On the other hand, the cell cycle of AGS cells is arrested in the G0/G1 phase, with CDKN1B and CDKN2B levels remaining unchanged after lycopene treatment. In normal conditions, both AGS and SGC7901 cells exhibit an amplification of cyclin E1 (CCNE1), an oncogene associated with uncontrolled cell proliferation [85]. After lycopene treatment, a reduction in CCNE1 levels occurs in both cancer cell lines, which suggest that it plays a key role in lycopene-mediated decrease in proliferation.

Like β-carotene, in the same research, lycopene was found to inhibit AtT20 cell proliferation and colony formation [67]. However, lycopene’s impact on the cell cycle was distinct. Lycopene induced cell cycle arrest in the G0/G1 phase while β-carotene induced cell cycle arrest in the M/G2 phase, evidencing that carotenoids can have different effects on cell proliferation in the same tumour cells.

#### 4.1.3. Role of Lutein in Cell Cycle Regulation and Antiproliferative Effects on Cancer Cells

Lutein is a prominent carotenoid found in vegetables, especially those showing green and yellow colours, which have the highest amounts. This carotenoid has anti-inflammatory action, and it has been used successfully in the treatment of diabetic retinopathy and cataracts, and enhancement of visual contrast [86]. Although the number of studies focused on the potential effects of lutein on tumoural cells is still low, it has been reported that lutein could decrease tumour cell proliferation. This carotenoid induces growth inhibition in BT-474, MDA-MB-453 and MDA-MB-231 human breast cancer cell lines [87]. This growth inhibition is primarily due to a G1 cell cycle arrest caused by the activation of tumour suppressor protein p53. Activated p53 induces cell cycle arrest, enabling DNA repair or apoptosis, to halt the proliferation of cells with severe DNA damage [88]. This occurs through the activation of its target genes that are involved in triggering cell cycle arrest or apoptosis. In addition to breast cancer, lutein also induces cell cycle arrest at the G0/G1 phase in lung adenocarcinoma A549 cell line in vitro, impedes tumour growth in mice and prolongs their survival [89].

#### 4.1.4. Role of Astaxanthin in Cell Cycle Regulation and Antiproliferative Effects on Cancer Cells

Found in marine organisms, astaxanthin induces cell cycle arrest of cancer cells, effectively restraining its proliferation. In the context of leukemia, this molecule inhibits the proliferation and decreases the viability of myeloid leukemia cell line K562 in dose- and time-dependent manners by causing G0/G1 cell cycle arrest [59]. This phenomenon involves the PPARγ signalling pathway and Keap1-Nrf2/EpRE/ARE signalling pathway, similar to what occurs with β-carotene treatment [59].

In addition, astaxanthin induces cell cycle arrest in the G0/G1 phase in human gastric adenocarcinoma cell lines KATO-III and SNU-1 by the down-regulation of the phosphorylated extracellular signal-regulated protein kinase (p-ERK) level, inhibiting the cyclin D1/CDK4 complex [90]. On the other hand, astaxanthin also increase p27^Kip1^, which induces the suppression of the cyclin E/CDK2 complex. These processes could cause cell cycle arrest at the G1/G0 phase in these cells [91]. This molecule blocks cell cycle progression at G0/G1 too in the SKBR3 breast cancer cell line, suppressing proliferation dose-dependently [92]. 

Astaxanthin inhibits cell growth of HCT-116 and HT-29 colon cancer cells too by inducing G2/M cell cycle arrest [93]. In this case, the cell cycle arrest is associated with increases in the expression levels of p21^Cip1/Waf1^, p27 and p53, as well as decreases in the levels of CDK4 and CDK6.

In breast cancer, astaxanthin administration reduces proliferation of cancer cells in 4T1 cell-injected mice [94]. Tumour tissues of treated mice present a significant reduction in mitotic cells.

#### 4.1.5. Role of Fucoxanthin in Cell Cycle Regulation and Antiproliferative Effects on Cancer Cells

Fucoxanthin is a natural pigment found in various types of brown and marine algae [5]. Its antitumour effects have been demonstrated in various cell lines in which, among other mechanisms, it has been able to induce cell cycle arrest and decrease proliferation.

Fucoxanthin’s impact on various leukemia cell lines, including those infected with human T-cell leukemia virus type 1 (HTLV-1) such as MT-2, MT-4, HUT-10, ED-40515(-) and Jurkat and K562, has been examined [95]. This carotenoid induces a significant G1 phase arrest across all these cell lines, caused by a reduction in the expression of crucial proteins involved in the G1/S transition, specifically cyclin D1, cyclin D2, CDK4 and CDK6, which are pivotal for advancing the cell cycle from G1 to S phase [95]. Additionally, treatment causes an up-regulation in the expression of growth arrest and DNA damage-inducible protein 45 α (GADD45α), a regulator known to inhibit the entry of cells into the S phase, in response to carotenoid treatment [96].

Furthermore, this compound has been tested in several human osteosarcoma cell lines, including Saos-2, MNNG and 143B, where it demonstrated the ability to inhibit osteosarcoma cell viability and induce G1 cell cycle arrest [97]. These effects were attributed to its capacity to reduce the expression of key proteins involved, specifically CDK4, CDK6 and cyclin E.

Fucoxanthin can exerts its antiproliferative effect by inducing cell cycle arrest in the G2/M in the gastric cancer cell line MGC-803 [98]. Notably, this impact is associated with the down-regulation of survivin and cyclin B1 through the Janus kinases/signal transducer and activator of transcription (JAK/STAT) signalling pathway. The JAK/STAT family are crucial components of various signal transduction pathways actively involved in cellular processes such as survival, proliferation, differentiation and apoptosis [99]. In addition, fucoxanthin has demonstrated its ability to inhibit the growth of other human gastric cell lines. This carotenoid induces cell cycle arrest at the S phase in SGC-7901 cells, while it causes G2/M phase arrest in BGC-823 cells [100]. These differences in phase arrest between the two cell lines might be attributed to their distinct cellular origins. On the one hand, BGC-823 cells are derived from primary gastric cancer tumour, while SGC-7901 cells are derived from metastatic lymph nodes around the stomach. These effects are due to the modulation of the expression of key proteins, including myeloid cell leukemia-1 (Mcl-1), STAT3 or p-STAT3, through JAK/STAT signalling pathway suppression [100].

Fucoxanthin also induces G1 phase arrest in HepG2 (liver cancer) and DU145 (prostate cancer) cell lines by the induction of the expression of GADD45α, GADD153 and proto-oncogene serine/threonine-protein kinase Pim-1 (PIM1) [101]. Moreover, fucoxanthin induces a marked increase in cytochrome P450 1A1 (CYP1A1) expression only in HepG2 cells, which could mean that carotenoids have effects on tumour cells mediated by both common and tumour-specific molecular mechanisms [102,103]. These findings led to the hypothesis that GADD45α, a p53-regulated gene known to interact with the products of two different p53-regulated genes, p21^Waf1/Cip1^ and proliferating cell nuclear antigen (PCNA), plays a role in fucoxanthin-induced G0/G1 arrest [103,104].

In the case of melanoma, fucoxanthin exerts antiproliferative effects on B16F10 melanoma cells, characterized by G1 cell cycle arrest [105]. This effect is due to the simultaneous down-regulation of cyclins D1 and D2, and CDK4 expression, while up-regulating p15^INK4B^ and p27^Kip1^ expression.

Moreover, at low doses (5 and 10 μM), fucoxanthin can induce G0/G1 growth arrest in human bladder carcinoma cells T24 [106]. In this case, the antiproliferative effect is produced by the up regulation of p21 followed by the down-regulation of CDK-2, CDK-4, cyclin D1 and cyclin E.

#### 4.1.6. Role of Capsanthin in Cell Cycle Regulation and Antiproliferative Effects on Cancer Cells

Capsanthin is a carotenoid compound found in red and orange peppers and it is responsible for giving these peppers their vibrant colour. While research on the specific effects of capsanthin on cancer is not as extensive as that for other carotenoids like β-carotene or lycopene, some studies have explored its potential benefits.

Capsanthin exhibits dose- and time-dependent inhibition of proliferation and reduced viability in the myeloid leukemia cell line K562 [59]. This effect is associated with G0/G1 cell cycle arrest and involves the activation of PPARγ signalling pathways as well as the Keap1-Nrf2/EpRE/ARE signalling pathway. 

In triple-negative breast cancer cell lines BT20, BT549, MDA-MB-468 and MDA-MB-231, but not in the normal human mammary epithelial line MCF-10A, capsanthin induces G1/S cell cycle arrest and inhibits tumour progression by suppressing the epigenetic silencing of p21 [107]. This silencing is mediated by enhancer of zeste homolog 2 (EZH2), a polycomb group protein implicated in cancer progression [108]. EZH2 overexpression often marks early precancerous changes in histologically normal mammary tissue and can lead to enhanced differentiation and proliferation [109].

#### 4.1.7. Role of Crocetin and Crocin in Cell Cycle Regulation and Antiproliferative Effects on Cancer Cells

Crocin (and the deglycosylated form called crocetin) is a carotenoid constituent of saffron that has also shown various pharmacological activities such as antioxidant, anticancer, memory improvement, antidepressant, cerebral, kidney, heart, skeletal muscle anti-ischemia, hypotensive, aphrodisiac, genoprotective and antidote activities [110]. Evidence suggests that crocin exhibits anticancer properties in female rats with NMU-induced breast cancer, effectively suppressing tumour growth [111]. Specifically, crocin counters the up-regulation of cyclin D1 expression observed in the NMU-treated group. This action leads to cell cycle arrest and a subsequent reduction in the proliferation rate of cancer cells. Also, this molecule produces a G1 arrest in colorectal cancer cells (SW-480 cell line) decreasing cell proliferation through the modulation of the Akt1 and JAK2/STAT3 signalling pathways [112]. Finally, in mice injected with AGS gastric cancer cell line, crocin reduces the proliferation of tumour cells [113].

### 4.2. Apoptosis Induction

Carotenoids are notable for their ability to induce tumour cell death by apoptosis, an essential mechanism that counteracts uncontrolled cell proliferation. Thanks to their ability to trigger programmed cell death pathways, these carotenoids show promise as agents that promote the elimination of cancer cells. A summary of preclinical studies on the induction of apoptosis in cancer cells by selected carotenoids is presented in Appendix A.

#### 4.2.1. Role of β-Carotene in Inducing Apoptosis in Cancer Cells 

The impact of β-carotene on programmed cell death, known as apoptosis, is a subject of significant research interest. In the context of breast cancer, β-carotene promotes an increase in cells in late apoptosis through JNK signalling in the triple-negative MDA-MB-231 cell line [57]. JNKs initiate apoptotic signalling by either increasing the expression of proapoptotic genes through the activation of specific transcription factors or directly modulating the functions of mitochondrial pro- and antiapoptotic proteins via distinct phosphorylation events [114]. The stimuli to activate this pathway could be ROS presence, certain growth factors or cytokines [115]. In addition, this molecule has been tested in others breast cancer cell lines with other phenotypes showing the same results. β-carotene induces also apoptotic effects in MCF-7 (luminal A), MDA-MB-235 (metastatic cells) and MDA-MB-231 (basal-like triple-negative) cell lines in a dose-dependent manner [71].

In another cancer context, β-carotene induces apoptosis in AtT20 cells (ACTH-secreting pituitary adenoma cells) [67]. In this case, apoptosis is promoted by the down-regulation of Skp2, which causes an increase in the amount of p27 [68,69].

#### 4.2.2. Role of Lycopene in Inducing Apoptosis in Cancer Cells

The induction of apoptosis in tumour cells by lycopene has been studied in a few different in vitro models. In breast cancer, lycopene treatment causes a significant increase in the apoptotic rate in MCF-7 (ER/PR positive), SK-BR-3 (HER2 positive) and MDA-MB-468 (triple-negative) breast cancer cell lines, accompanied by the cleavage of poly ADP-ribose polymerase (PARP) [81,116,117]. This enzyme participates in repair of DNA damage by adding poly (ADP ribose) polymers in response to a variety of cellular stresses and its cleavage by caspases is considered a hallmark of apoptosis [118]. The apoptotic effects induced by lycopene vary depending on the receptor status. Specifically, lycopene induces marked apoptosis in triple-negative breast cancer cells, whereas it has a minor impact on hormone receptor- and HER2-positive cells [116].

Telomerase activity, a common feature in various cancer cells, is associated with resistance to apoptosis through multiple mechanisms [119,120]. On leukemia cancer cells, lycopene exhibits a dose- and time-dependent inhibition of telomerase activity in K562 cells, which positively correlates with induction of apoptosis [121]. 

Regarding colon cancer, lycopene induces apoptosis in HT-29 cell line due to a decrease in the expression levels of procaspases-8, -3 and -9, as well as PARP-1 and B-cell lymphoma 2 protein (Bcl-2), accompanied by an enhancement in Bcl-2-associated X-protein (Bax) expression [81,122]. The Bcl-2 family includes proteins that regulate programmed cell death by either inhibiting (antiapoptotic) or inducing (proapoptotic) apoptosis [123]. Bax protein, a member of this family, mediates mitochondrial membrane permeabilization during apoptosis, which leads to the release of cytochrome-c and other proapoptotic factors from the mitochondria, triggering caspase activation [123].

Lycopene also has the potential to trigger apoptosis in PC3 cells (prostate cancer cell line) by up-regulating miR-let-7f-1 expression and concurrent inhibition of Akt2, a pivotal component of the PI3K signalling pathway [124]. Akt2 is highly expressed in many human cancers and is closely associated with cancer cell metabolism, proliferation, cell survival, metastasis, angiogenesis and drug resistance [125,126].

In a study with an oesophageal cancer-induced model in F344 rats, the lycopene treatment resulted to enhance apoptosis by increasing the protein expression levels of PPARγ and caspase-3 while reducing inflammatory cytokines by decreasing the protein expression of NF-κB and cyclooxygenase-2 (COX-2) in oesophageal tissue [127]. Phosphorylated PPARγ interacted with p65, inhibiting NF-κB activity, which in turn affected the expression of tumour-associated genes and transcription factors, ultimately suppressing cytokine production [128]. COX-2, highly expressed in malignant tumours, plays a role in promoting cell proliferation and inhibiting apoptosis [129].

In ACTH-secreting pituitary adenoma cells AtT20, lycopene also induces apoptosis by up-regulating p27 protein, which happens with this cell line when treated with β-carotene [67]. This could suggest that different carotenoids have similar anticancer properties in the same tumour cells. On the other hand, the intensity of the effect varies if we treat different cancer types with the same carotenoid. For example, the apoptosis inducing activity of lycopene in different cell lines of various cancer types (PC3, human prostate cancer; MCF-7, human breast adenocarcinoma; HeLa, human cervical cancer; A431, human epidermoid carcinoma; HepG2, hepatocellular carcinoma; A549, human lung adenocarcinoma) has been investigated, evidencing that this carotenoid induce apoptosis with ROS generation as an intermediate step, but the effect is much greater in PC3, MCF-7 and HeLa cell lines [130].

#### 4.2.3. Role of Lutein in Inducing Apoptosis in Cancer Cells

Within the intricate landscape of carotenoid-induced apoptosis, the role of lutein has been moderately explored. In the case of breast cancer, lutein exhibits apoptosis induction in various human breast cancer cell lines (BT474, CRL-3247, MCF-7, MDA-MB-231, MDA-MB-453, MDA-MB-468, HTB-26, HTB-131) by increasing intracellular ROS levels, p53 signalling activation and up-regulation of cellular heat shock protein 60 (HSP60), which leads to a decrease in Bcl-2 expression and a concurrent increase in Bax protein expression [87]. Notably, lutein induces apoptotic cell death through a caspase-independent mitochondrial pathway, as caspase-3 cleavage is not activated [87]. Unlike many HSPs that typically have prosurvival roles, cytosolic HSP60 can have dual effects, being either proapoptotic or prosurvival, contingent on its interactions with caspase-3 [131]. In addition, the proapoptotic effect of lutein occurs specifically in tumour cells and not in primary normal human mammary epithelial cells [132]. However, caspase-3-mediated apoptosis with intracellular ROS reduction after lutein treatment has also been demonstrated in MCF-7 and MDA-MB-231 cell lines [133]. The proapoptotic effect of lutein has also been observed in female BALB/c mice [134]. Tumour cells in these mice show an increased expression of proapoptotic genes, such as p53 and Bax, and a decreased expression of the antiapoptotic gene Bcl-2.

The specificity of lutein in the induction of apoptosis in tumour cells has been described also in lung cancer context. Lutein induces apoptosis by modulating the PI3K/Akt signalling pathway in A549 and HCC827 lung cancer cells, whereas it does not affect BEAS-2B normal cells [135].

Regarding gastric cancer, lutein induces apoptosis in AGS, MKN-74, MKN-1 and SNU-668 cell lines by elevating levels of ROS through NADPH oxidase activation [136]. The increase in ROS triggers the activation of NF-κB, up-regulation of proapoptotic factors such as Bax, caspase-3 cleavage and DNA fragmentation, as well as a decrease in the Bcl-2 protein.

#### 4.2.4. Role of Astaxanthin in Inducing Apoptosis in Cancer Cells

In terms of its ability to trigger apoptosis, astaxanthin is emerging as a relatively underexplored carotenoid. In a mouse model injected with B16F10 skin melanoma cells that progressed to lung metastatic melanoma, astaxanthin induces apoptosis in melanoma cells within the lung by inhibiting key proteins such as Bcl-2, cyclins D1 and E, NF-κB, ERK, mitogen-activated protein kinase kinase (MEK) and matrix metalloproteinases 1 and 9 (MMP-1 and -9), while increasing cleaved caspase-3 and -9, ataxia telangiectasia-mutated kinase (ATM) and p21 [137]. Dysregulated cyclin E activity and overactive MEK were also implicated in cancer progression [138]. Therefore, the overactivity of MEK is a common occurrence in certain cancers. When MEK is inhibited, it helps to control tumour growth and facilitates apoptosis [139]. The MMPs family of proteins is closely associated with the degradation of various extracellular matrices in normal physiological processes. MMPs are known to play roles in tumour adhesion and dispersion, the inactivation of cytokines and chemokines, the release of apoptotic ligands, angiogenesis and the cleavage of cell surface receptors [140].

Astaxanthin inhibits key signalling pathways such as NF-κB and wingless/Int-1 (Wnt) in a hamster model of oral cancer through the down-regulation of critical regulatory enzymes, including the inhibitor of nuclear factor kappa B and glycogen synthase kinase-3β GSK-3β [141]. Additionally, this carotenoid induces apoptosis through the caspase-mediated mitochondrial pathway, leading to the release of Smac/Diablo and cytochrome-c into the cytosol and the cleavage of PARP.

Regarding colon cancer, astaxanthin induces apoptosis in HCT-116 and HT-29 human colon cancer cells through the activation of caspase-3 and PARP [93]. In the case of the HT-29 cells, astaxanthin-mediated apoptosis involves the induction of ROS production [142]. The apoptosis with caspase-3 activation also was described in the LS-80 human colon cancer cell line [143].

This carotenoid has also been tested in breast cancer models. Astaxanthin induces apoptosis in the SKBR3 cell line by decreasing the expression level of mutant p53 and cleaving a PARP-1 fragment [92]. Furthermore, it induces the activation of Bax/Bcl-2, cleaved caspase-3 and cleaved caspase-9 as well as the phosphorylation of ERK1/2, JNK and p38. Intracellular ROS decrease in a dose-dependent manner after treatment, accompanied by inhibition of SOD1 expression and up-regulation of SOD2. It is well known that the balance between both enzymes is a critical mechanism of stress-mediated regulation [144], so astaxanthin plays an important role in the induction of ROS imbalance in cancer cells.

#### 4.2.5. Role of Fucoxanthin in Inducing Apoptosis in Cancer Cells

Fucoxanthin, one of the most studied carotenoids, has numerous studies supporting its ability to induce apoptosis in tumour cells. Its prevalence in dietary sources underlines its importance in both biomedicine and human nutrition.

In a mouse model of lung cancer induced with benzo(a)pyrene, fucoxanthin plays a role in the chemoprevention of lung cancer [145]. The carotenoid increases the apoptotic rate of cancer cells by up-regulating caspase-9 and -3 while down-regulating the antiapoptotic protein Bcl-2. Additionally, fucoxanthin aids in the recovery of damaged tissue treated with benzo(a)pyrene and decreases PCNA expression.

Fucoxanthin induces apoptosis in various leukemia cell lines, including those infected with HTLV-1, such as MT-2, MT-4, HUT-10 and ED-40515(-) and Jurkat and K562 [95]. The compound reduces the expression of Bcl-2, mitochondrial X-linked inhibitor of apoptosis protein (XIAP), cellular inhibitor of apoptosis 2 (cIAP2) and survivin, thereby activating caspase-3, -8 and -9. Specifically, fucoxanthin promotes caspase-3- and -9-dependent apoptosis by down-regulating antiapoptotic proteins XIAP and cIAP2 expression. Additionally, fucoxanthin inhibits IκBα phosphorylation, leading to the inactivation of NF-κB and activator protein 1. 

Fucoxanthin induces apoptosis in canine mammary tumour cells CMT-U27 by activating caspases-3, -7, and -8, and PARP [146]. Caspase-8, specifically, plays a significant role in promoting apoptosis, particularly through extrinsic pathways involving the Fas-associated death domain [147].

A study investigated human cervical cancer cell lines (HeLa, SiHa and CaSki) treated with fucoxanthin, resulting in apoptosis induction in these cells [148]. Furthermore, the research suggests that fucoxanthin’s apoptotic effects are linked to the inhibition of the PI3K/Akt and NF-κB signalling pathways.

Fucoxanthin induces apoptosis, the loss of mitochondrial membrane potential (ΔΨm) and ultrastructural alterations in glioblastoma cell line GBM1 [149]. The initial phase of the intrinsic apoptosis pathway is characterized by the permeabilization of the mitochondrial membrane, leading to a decline in membrane potential [150]. This event triggers the release of proapoptotic proteins such as apoptosis-inducing factor and cytochrome-c into the cytosol. On the other hand, fucoxanthin also induces apoptosis in U251 human glioma cells by initiating oxidative damage through ROS and disrupting the functionality of MAPKs and the PI3K-Akt pathways [151].

The antiosteosarcoma potential of fucoxanthin was also assessed across multiple osteosarcoma cell lines, including Saos-2, MNNG, 143B and LM8 [97]. Fucoxanthin exhibits apoptosis-inducing properties by down-regulating the expression of survivin, XIAP, Bcl-2 and Bcl-xL in these cell lines. This apoptotic process is mediated by the activation of caspases-3, -8 and -9.

Fucoxanthin has been tested in bladder cancer too. The carotenoid triggers apoptosis in T24 human bladder cancer cells by disrupting the mortalin-p53 complex and restoring the nuclear function of p53, which acts as a tumour suppressor [106].

In the case of gastric cancer, fucoxanthin induces apoptosis through the JAK/STAT signal pathway on human gastric adenocarcinoma SGC-7901 and BGC-823 cells [100]. Specifically, the mechanism involves the down-regulation of Mcl-1, STAT3 and p-STAT3.

#### 4.2.6. Role of Capsanthin in Inducing Apoptosis in Cancer Cells

The induction of apoptosis in cancer cells by capsanthin has been studied very little. In the breast cancer context, this carotenoid exhibits a dose-dependent induction of apoptosis and a reduction in cell viability of MCF-7 cells [152]. This effect is associated with the generation of oxidative stress and a decrease in mitochondrial membrane potential, accompanied by a reduction in glutathione and catalase levels. Furthermore, capsanthin induced apoptosis and increased ROS in MDA-MB-231 cell line [153,154].

In addition, this carotenoid has been tested in a leukemia model. Specifically, capsanthin induces apoptosis in myeloid leukemia cells K562 by up-regulating PPARγ, p21 and Nrf2 [59].

#### 4.2.7. Role of Crocetin and Crocin in Inducing Apoptosis in Cancer Cells

Crocetin and crocin, two apocarotenoids commonly found in the diet, have gained significant attention in biomedicine for their potential role in inducing apoptosis in cancer cells. In the context of gastric cancer, crocetin induces apoptosis in gastric adenocarcinoma AGS cells by suppressing Bcl-2 and increasing Bax expression, with no observed toxic effects in normal human fibroblast HFSF-PI3 cells [155,156]. This highlights the potential selectivity of carotenoids against tumour cells.

Crocetin treatment suppresses fatty acid synthase (FASN) expression in glioblastoma cell lines (U251, U87MG, U373 and U138), activating apoptosis pathways, as evidenced by the presence of cleaved caspase-3 bands [157]. FASN expression, typically correlated with glioblastoma aggressiveness, is a key player in lipid synthesis whose inhibition induced programmed cell death in cancer cells [158].

In retinoblastoma, crocin induces apoptosis in Y79 and WERI-RB-1 cell lines with a significant reduction in the expression and stability of the proto-oncogene N-myc (MYCN) protein [159]. MYCN, which is part of the MYC family and is a transcription factor with a basic helix–loop–helix domain, plays a critical role in gene expression, protein synthesis and essential cellular processes [160]. In addition, MYCN plays a critical role in cancer onset, progression and invasive tendency, making it a key factor in the maintenance of tumourigenic states in various types of cancer [161]. Due to the versatility of MYCN, the key activities and mechanisms by which MYCN drives cancer remain unclear.

Crocetin also induces cell apoptosis in oesophageal squamous cell carcinoma cell line KYSE-150 [162]. In this case, the apocarotenoid inhibits the activation of PI3K/Akt, ERK1/2 and p38, and up-regulated p53 and p21 tumour suppressor proteins. These regulations ultimately trigger the mitochondrial-mediated apoptosis pathway with the disruption of MMPs, increased levels of Bax and cleaved caspase-3, and decreased levels of Bcl-2. 

Proapototic activity of crocin occurs even in chemoresistant tumour cells. In OV2008 (chemosensitive) and C13 (chemoresistant) human cervical cancer cell lines, crocin induces apoptosis by up-regulating the expression of proapoptotic factors Bax and p53, while down-regulating the expression of the antiapoptotic factor Bcl-2 and its upstream regulator, miR-365 [163]. The effect described above underscore the value of carotenoids as therapeutic agents against cancer, given that chemoresistance remains a major obstacle to the effective treatment of ovarian cancer, among others [164]. Continuing with cervical cancer, p53-mediated apoptosis has been observed in the HeLa cell line, this time with crocetin instead of crocin [165]. This mechanism following crocetin treatment has also been described in A549 lung and SKOV3 ovarian cancer cell lines [165].

Regarding leukemia, crocetin induces apoptosis in HL-60 cell line whereas normal human polymorphonuclear cells shows no significant toxicity [166]. Therefore, crocetin treatment results in a significant increase in the expressions of caspases-3 and -9, as well as an elevated Bax/Bcl-2 ratio, indicating the activation of the intrinsic apoptotic pathway. 

Crocin, when tested on human breast cancer cells (MDA-MB-468) and normal breast epithelial cells (MCF10-A), selectively induces apoptosis in the breast cancer cells without affecting the normal epithelial cells [167]. Notably, crocin treatment led to an increase in the Bax/Bcl2 ratio and a reduction in the expression of antiapoptotic HSP27, 70 and 90, signifying a proapoptotic change in the cellular environment. These HSPs are intricately linked to pivotal signalling pathways involved in stress responses and apoptosis regulation, thus acting to inhibit apoptosis while supporting cell survival, proliferation, or differentiation [168]. In another study involving MCF-7 and MDA-MB-231 breast cancer cell lines, crocin triggers a series of events culminating in apoptosis induction [169]. Specifically, crocin initiates the production of ROS, leading to an increased expression and nuclear translocation of forkhead box transcription factor class O 3a (FOXO3a), which up-regulates the expression of Bim (encodes Bcl-2-like 11 protein) and PTEN (encodes phosphatase and tensin homolog protein) genes and activates caspase-3. The FOXO protein family comprises various members, among them FOXO3a, which can activate a range of target genes in response to oxidative and other stress stimuli [170]. These genes encompass Bim and TRAIL (encodes tumour necrosis factor-related apoptosis-inducing ligand protein), involved in apoptosis induction; p27, p21 and cyclin D, which play roles in cell cycle arrest; and PTEN, which regulates both apoptosis and cell cycle arrest [171,172,173].

Finally, crocetin has been demonstrated apoptotic potential in colon cancer cell lines, including HCT-116 and HT-29 [174]. Crocetin triggers a cascade of events involving p53, which transactivates Bax and up-regulates p53-induced death domain protein (PIDD). PIDD subsequently cleaves and activates BH3-interacting domain death agonist (BID) through caspase-2. Both Bax and BID converge at the mitochondria, disrupting the transmembrane potential and ultimately leading to caspase-3 and -9 mediated apoptosis in these colon cancer cells.

### 4.3. Inhibition of Metastasis and Antiangiogenic Effect

Metastasis is an intricate process by which cancer cells detach from the primary tumour, travel through the bloodstream or lymphatic system and colonize distant organs, leading to the formation of secondary tumours [1]. It represents a major hallmark of malignancy and stands as a primary contributor to the high mortality rates associated with advanced-stage cancers [2].

At the heart of this malignant process is angiogenesis, the intricate mechanism responsible for forming new blood vessels. While angiogenesis plays crucial roles in normal physiological processes, such as embryonic development, wound healing and tissue repair, it becomes a hallmark of malignancy when driven by cancer cells [175]. Cancer cells, driven by their insatiable hunger for nutrients and oxygen, manipulate angiogenesis to generate a network of blood vessels within the tumour microenvironment [176,177]. These newly formed vessels supply the growing tumour with vital resources, allowing it to proliferate and facilitating the dissemination of cancer cells to distant sites.

In the ongoing research of novel strategies to combat cancer, carotenoids have demonstrated antimetastatic and antiangiogenic properties against cancer cells. By intervening at various points along the metastatic cascade, these antioxidant molecules provide multiple strategies to impede the spread of cancer cells and reduce the devastating impact of metastatic disease. A summary of preclinical studies on the antimetastatic and antiangiogenic effect of selected carotenoids is presented in Appendix A.

#### 4.3.1. Inhibition of Metastasis and Antiangiogenic Effects of β-Carotene in Cancer Cells

Regarding the antimetastatic effect of β-carotene, this carotenoid exerts an inhibitory effect on MAPK-mediated MMP-10 expression and cell invasion by increasing PPAR-γ-mediated catalase expression and reducing ROS levels in *Helicobacter pylori* (*H. pylori*)-infected gastric adenocarcinoma cells (AGS cell line) [178]. MMPs, key molecules of cancer invasion and metastasis, degrade the extracellular matrix and cell–cell adhesion molecule [129]. 

In the neuroblastoma context, β-carotene can suppress the metastasis of malignant neuroblastoma cells both in vitro and in vivo. The carotene attenuates the migratory and invasive capabilities of SK-N-BE(2)C neuroblastoma cell line in vitro and in metastasis observed in immunodeficient nude mice injected with SK-N-BE(2)C cells [179]. Specifically, β-carotene treatment suppresses the activity and expression of MMP-2 and -9 and hypoxia-inducible factor-1α (HIF-1α), which are crucial proteins involved in cell invasion and metastasis [180]. HIF-1α is responsible for activating the transcription of various genes, including vascular endothelial growth factor (VEGF) and glucose transporter 1 (GLUT1). VEGF plays a central role in angiogenesis and tumour progression and its expression is crucial for tumour growth, invasion and metastasis [181]. Similarly, GLUT1, an integral glycoprotein, facilitates glucose transport during hypoxia [182].

β-carotene also inhibits both metastasis and angiogenesis in in vivo and in vitro metastatic melanoma models performed with the B16F10 cell line and C57BL/6 injected mice [183,184]. The treatment with this molecule down-regulates the expression of MMP-2 and -9, prolyl hydroxylase and lysyl oxidase and up-regulates the expression of tissue inhibitor of metalloproteinase (TIMP) 1 and 2. TIMP-1 and -2 can form a complex with pro-MMP-9 and -2, respectively, leading to inhibition of the activity of MMPs [185]. Moreover, prolyl hydroxylase and lysyl oxidase are key enzymes required for the stabilization of collagen, a process required for angiogenesis [186].

#### 4.3.2. Inhibition of Metastasis and Antiangiogenic Effects of Lycopene in Cancer Cells

Delving deeper into the therapeutic potential of lycopene, we found that this molecule produces an inhibitory force against tumour cell invasion in in vitro experiments with human head and neck squamous cell carcinoma (HNSCC) cell lines FaDu and Cal27 [187]. The inhibition of the aggressive behaviour of HNSCC cell lines is associated with decreased expression levels of p-Akt and p-ERK.

In a study involving HT-29 human colon cancer cell line, lycopene led to a reduction in cell migration [122]. This effect was accompanied by a shift towards a less invasive phenotype, as indicated by the up-regulation of the epithelial marker E-cadherin, which acts as a tumour suppressor protein and its diminished expression often coincides with the process of epithelial–mesenchymal transition (EMT), a common event in tumour metastasis [188]. Additionally, the carotene down-regulated critical molecules associated with cancer cell invasion, including Akt, NF-κB, pro-MMP-2 and active MMP-9.

The EMT is a pivotal cellular process that holds significant implications in the realm of cancer metastasis. EMT represents a fundamental transformation in cell behaviour, where epithelial cells lose their characteristic adhesion properties and acquire a more motile, mesenchymal-like phenotype [189]. This phenotypic switch is not only a central feature in embryonic development and tissue repair but also plays a critical role in the metastatic cascade of cancer cells. During EMT, cancer cells undergo a series of molecular changes, including the down-regulation of cell adhesion molecules like E-cadherin and the up-regulation of mesenchymal markers such as N-cadherin, vimentin and various transcription factors [176]. These alterations provide cancer cells with enhanced migratory and invasive capabilities, allowing them to break free from the primary tumour site, invade surrounding tissues, enter the bloodstream or lymphatic system and eventually colonize distant organs [189]. Consequently, EMT serves as a bridge that links primary tumour growth to the initiation of metastasis, highlighting its significance as a potential target for therapeutic interventions aimed at impeding cancer’s deadliest stage. Understanding the intricate molecular mechanisms governing EMT in the context of metastasis is essential for developing novel strategies to combat cancer progression and improve patient outcomes.

Lycopene also inhibits adhesion of prostate cancer cell lines PC3 and DU145 to Matrigel™ [190]. This suggests its potential to impede the initial stages of cancer cell interaction with the extracellular matrix.

In an aggressive hepatocarcinoma in vitro model utilizing the SK-Hep-1 cell line, lycopene exhibits the capacity to significantly diminish metastasis-related traits, including cell migration and invasion [191,192]. This antimetastatic action of lycopene is mediated by the up-regulation of the expression of Nm23-H1, a nucleoside diphosphate kinase-A (NDPK-A), which constitutes the pioneering identified metastasis suppressor protein [193].

#### 4.3.3. Inhibition of Metastasis and Antiangiogenic Effects of Lutein in Cancer Cells

The antimetastatic potential of lutein remains poorly studied. It has recently been described that lutein can suppress the migration and invasion of pancreatic adenocarcinoma cells PANC-1 by targeting Bcl2-associated athanogene 3 and modulating cholesterol homeostasis [194]. The dysregulation of cholesterol homeostasis at the cellular level is known to contribute to tumour progression [195]. 

In addition, lutein inhibits invasion and migration of breast cancer cell lines MDA-MB-157 and MCF-7 via down-regulation of hairy and enhancer of split 1 protein [196]. This protein plays a role in controlling cell proliferation and differentiation in embryonic development, the maintenance of stem cells and the progression and survival of tumour cells [197].

#### 4.3.4. Inhibition of Metastasis and Antiangiogenic Effects of Astaxanthin in Cancer Cells

While this carotenoid is better known for its antioxidant properties, recent studies have unveiled its intriguing potential as an antimetastatic agent. In colon cancer area, astaxanthin suppresses the metastatic capacity of colon cancer cell line HCT116 in vitro, and CT26 cells injected in BALB/c nu/nu mice, by inhibiting the MYC-mediated down-regulation of microRNA (miR)-29a-3p and miR-200a [198]. In addition, natural compounds inhibit tumour invasion in an induced rat model by modulating the expressions of NF-κB, COX-2, MMPs-2/9, Akt and ERK-2 [199]. 

Regarding breast cancer, astaxanthin increases the activation of metastasis, suppressing mammary serine protease inhibitor, KAI1 (CD82), breast cancer metastasis suppressor 1 and mitogen-activated protein kinase kinase 4 on T47D cells [200]. All of them are proteins that block metastasis without inhibiting the formation of primary tumours, but the mechanisms are still unknown [201].

In gastric cancer, astaxanthin reduces cell adhesion and migration of *H. pylori*-stimulated AGS cells [202]. Specifically, the carotenoid supresses JAK1/STAT3 activation and blocks integrin α5, via AG490 (JAK/STAT inhibitor) and integrin α5β1 antagonist K34C. 

Finally, astaxanthin plays an antimetastatic role in glioblastoma. This molecule decreases migration and invasion abilities of A172 cells in a time- and dose-dependent manner with the down-regulation of MMP-2 and MMP-9 [203].

#### 4.3.5. Inhibition of Metastasis and Antiangiogenic Effects of Fucoxanthin in Cancer Cells

While fucoxanthin is better known for its antioxidant properties, recent studies have described its antimetastatic potential. In breast cancer, fucoxanthin hinders metastasis in MCF-7 and 4T1 cell lines by preventing circulating tumour cells (CTCs) from adhering to endothelial cells and inhibiting their transendothelial migration [204]. This is achieved by suppressing the expression of cell adhesion molecules induced by TNF-α on endothelial cells, regulating the NF-κB pathway and targeting the EMT, PI3K/Akt and focal adhesion kinase/Paxillin signalling pathways. The adhesion of CTCs to vascular endothelial cells and their subsequent passage across these cells are essential stages in the formation of micrometastatic sites distant from the primary tumour [205]. Consequently, disrupting the adhesion of CTCs to endothelial cells and their transendothelial migration holds significant potential in effectively preventing cancer metastasis. 

In MDA-MB-231 cell line and human lymphatic endothelial cells, fucoxanthin inhibits tumour-related lymphangiogenesis, decreasing levels of VEGF-C, VEGF receptor-3, NF-κB, phospho-Akt and phospho-PI3K [206]. Fucoxanthin also decreases micro-lymphatic vascular density in a MDA-MB-231 nude mouse model [206]. Lymphangiogenesis, a key process in cancer metastasis, involves the growth of new lymphatic vessels [207]. This phenomenon is characterized by the proliferation and migration of lymphatic endothelial cells, driven by the secretion of lymphangiogenic factors from various elements within the tumour microenvironment, including tumour cells and inflammatory cells. 

Fucoxanthin also exerts antiangiogenic activity on human umbilical vein endothelial cells and CMT-U27 canine mammary tumour cells by promoting angiopoietin 2 (Ang2) expression [146]. Ang2 plays a key role in promoting angiogenesis and stability in vascular physiology and the imbalance of its expression is an important condition for the occurrence and development of cancer [208]. Finally, there is also evidence of fucoxanthin decreased migration and invasion of glioblastoma GBM1 cells [149].

#### 4.3.6. Inhibition of Metastasis and Antiangiogenic Effects of Capsanthin in Cancer Cells

Although there is evidence to suggest that capsanthin can inhibit cell proliferation and promote apoptosis in several cancer cell lines, as we have seen previously in this review, research specifically addressing its role in metastasis and angiogenesis remains unknown. Given the importance of metastasis in cancer progression and the pivotal role of angiogenesis in tumour growth and dissemination, studying the antimetastatic potential of capsanthin could provide valuable insights into its broader therapeutic utility against cancer.

#### 4.3.7. Inhibition of Metastasis and Antiangiogenic Effects of Crocetin and Crocin in Cancer Cells

Crocin and crocetin show promise in cancer research by slowing tumour growth and triggering cell death in various cancer cell lines, as discussed above. Investigating whether these compounds can also interfere with the spread of cancer through processes such as cell migration, invasion and angiogenesis could advance our understanding of their potential use in biomedicine.

Regarding gastric cancer, crocetin suppresses angiogenesis and metastasis in NCI–N87 and Hs-746T cell lines through inhibiting the sonic hedgehog (SHH) signalling pathway [209]. Being newly recognized as a proangiogenic factor, SHH engages with its receptor patched, initiating the release of smoothened, which in turn orchestrates the downstream activation of glioma-associated oncogene homolog 1 proteins (GLI1), which induces the transcription of others downstream target genes [210]. This intricate signalling cascade governs an array of target genes associated with cell growth and angiogenesis. However, another study on gastric cancer demonstrated that crocin inhibits the EMT, migration and invasion of cancer cells obtained from biopsies of gastric cancer patients, through the miR-320/Krüppel-like factor 5 (KLF5)/HIF-1α signalling pathway [211]. In this context, KLF5 acts as a pivotal regulator of multiple cellular processes, modulating gene expression in response to environmental signals to influence cell proliferation, apoptosis, migration, differentiation and stem cell characteristics [212].

In breast cancer, crocin suppresses metastatic breast cancer progression via VEGF and MMP-9 down-regulations both in vitro with 4T1 cell line and in vivo with BALB/c mice [213]. In addition, in this model it has been described that crocin causes the down-regulation of the expression of Wnt/β-catenin target genes in tumours [214]. The dysregulation of the Wnt/β-catenin pathway, a family of proteins crucial for embryonic development and adult tissue maintenance, is frequently associated with severe diseases, notably cancer [215]. In MDA-MB-231 cells, crocetin inhibits cellular invasiveness by down-regulating the expression of MMP-2 and -9 [216,217].

In the context of melanoma, crocin decreases cell migration and invasion of B16F10 cell line, as well as reducing metastasis in C57BL/6 mice injected with B16F-10 cells [218]. This effect occurs through the down-regulation of MMP-2 and -9, ERK-2, K-ras and VEGF expression. 

Finally, crocin inhibits angiogenesis and metastasis in colorectal cancer cell lines HT-29 and Caco-2, blocking the TNF-α/NF-κB/VEGF pathways [219]. In the case of crocetin, it suppresses the growth and migration in HCT-116 human colorectal cancer cells by activating the p-38 MAPK signalling pathway [220].

### 4.4. Other Effects

In addition to their well-known effects such as antiproliferative action, the induction of apoptosis and reduction in metastatic capacity, carotenoids may also exert less common but significant effects. Specifically, these antioxidant compounds can induce cell death in cancer cells by autophagy or necroptosis, induce cell differentiation, enhance gap junctional communication or mitigate multidrug resistance. A summary of preclinical studies on the less common effects of selected carotenoids on cancer is presented in Appendix A.

#### 4.4.1. Autophagy and Necroptosis

While apoptosis remains the most studied mechanism by which carotenoids exert their cell death mechanism in cancer cells, emerging research highlights their involvement in other cell death pathways, such as autophagy and necroptosis (hybrid cell death pathway between apoptosis and necrosis). These alternative modes of cellular death add to the complexity of carotenoid-mediated responses in cancer cells.

Autophagy is a tightly regulated cellular process involving the degradation and recycling of damaged organelles and proteins. Recent studies have unveiled the ability of certain carotenoids to induce autophagy in cancer cells. For instance, fucoxanthin exerted autophagy-dependent cytotoxic effect in HeLa cells via inhibition of the Akt/mTOR signalling pathway [221]. In addition, this carotenoid induced autophagy in the nasopharyngeal carcinoma cell line NPC [222]. Furthermore, crocin treatment in cervical cancer SiHa cells also induces autophagic cell death through the activation of PI3K/Akt in vitro and in female BALB/c nude mice injected with SiHa cells [223]. 

However, autophagy has dual roles in cancer, acting as both a tumour suppressor and promoter, depending on context [224]. In its protumoural role, it can enhance cell migration in certain cell types [225]. In human cutaneous squamous cell carcinoma Colo-16 cell line where autophagy plays a protumoural role, lycopene inhibits this mechanism by up-regulating zonula occludens-1 expression and downregulating claudin-1 expression through the activation of ERK, JNK and mammalian target of rapamycin complex 1 [226]. 

On the other hand, necroptosis, a regulated form of necrosis, remains relatively scarce. To date, only one case of necroptosis induced by a carotenoid has been described. This is the case of astaxanthin, which induces necroptotic cell death by increasing NADPH oxidase activity, ROS levels and lactate dehydrogenase release as well as activating necroptosis-regulating protein, receptor-interacting protein 1 and 3 and mixed lineage kinase domain-like protein in gastric cancer AGS cells [227].

#### 4.4.2. Induction of Cell Differentiation

Differentiation therapy is an innovative approach in cancer treatment that focuses on transforming cancer stem cells, which can self-renew and resist traditional treatments, into more mature and less aggressive cells. By inducing these cancer stem cells to differentiate into non-stem cancer cells, their capacity for uncontrolled growth is diminished. This strategy offers new hope for more effective and less toxic treatments, especially in cancers where cancer stem cells play a pivotal role in tumour progression and recurrence.

By driving cancer cells into more specialized forms, carotenoids can arrest uncontrolled proliferation. Lycopene induces the differentiation of the HL-60 promyelocytic leukemia cell line into the monocyte/macrophage lineage, decreasing stemness by differentiating cancer cells [83]. Furthermore, β-carotene inhibits cancer cell stemness in neuroblastoma cells SK-N-BE(2)C and SH-SY5Y by regulating differentiation-related miRNAs and inducing neuronal differentiation [228,229]. In the case of crocetin, it reduces the levels of mesenchymal markers (CD44, CD90, CXCR4 and OCT3/4) and induces an increase in neuronal markers (βIII-tubulin and neurofilaments) in U251 glioma cells, changing its morphology from a polygonal shape to a thinner and elongated cell body shape [157].

#### 4.4.3. Enhancement of Gap Junctional Communication

Gap junction (GJs) channels are formed by integral membrane proteins called connexins (Cxs). Gap junctional intercellular communication (GJIC) significantly influences cancer progression, impacting malignancy, invasiveness, drug response and even the side effects of radiotherapy [230]. Generally, tumour cells are deficient in communicating junctions [231]. This reduction or loss of GJs between normal and cancer cells heightens malignancy and metastatic potential, while up-regulating GJs numbers can reduce drug resistance [232]. Moreover, GJIC enhances the cytotoxicity of chemotherapeutic agents, contributing to its role in cancer management [232]. However, GJIC is involved in the side effects of radiotherapy, allowing harmful substances generated during treatments, like ROS in chemotherapy, to affect neighbouring cells through GJs [232]. For this reason, in recent years Cxs have been classified as conditional tumour suppressors that can modulate cellular proliferation and metastasis [230].

In this context, GJIC can be regulated by carotenoids. Fucoxanthin induces cell cycle arrest and apoptosis in SK-Hep-1 human hepatoma cells through the up-regulation of both Cx43 and Cx32 expression, increment of intracellular Ca^+2^ levels and enhancement of GJIC [233]. In ACTH-secreting pituitary adenoma cells AtT-20, the treatment with β-carotene or lycopene causes cell death by apoptosis and produces a reduction in total Cx-43 levels but increases phosphorylated Cx-43 (pCx-43) levels [67]. Recent research indicates that pCx43 plays a role in regulating cell growth and death and can influence gene transcription or modulate proteins that control the cell cycle, such as Skp2 and p27^Kip1^ [234]. In contrast, lycopene inhibits cell proliferation by inducing cell cycle alterations on the human prostate cell line DU145, without any impact on Cx43 expression [235].

The effect of carotenoids on GJIC in tumour cells is not clear. Furthermore, there is increasing doubt about whether GJIC plays a pro- or antitumoural role. In this context, it seems that there are different signalling pathways that affect cell junctions and that can also be modulated through carotenoids and produce cell death of tumour cells. However, these mechanisms must be further investigated and studies scaled to different carotenoids and types of cancer must be conducted.

#### 4.4.4. Multidrug Resistance

Despite the availability of various cancer treatment methods such as radiation therapy, surgery, immunotherapy, endocrine therapy and gene therapy, chemotherapy remains the predominant approach. However, an alarming statistic reveals that over 90% of cancer-related deaths are attributed to the development of drug resistance [236]. Multidrug resistance (MDR) in cancer cells, especially during chemotherapy, can be linked to several intricate mechanisms. These include the enhanced efflux of drugs, genetic factors such as gene mutations, amplifications and epigenetic alterations, the influence of growth factors, elevated DNA repair capacity and increased metabolism of xenobiotics [237]. 

The activity of ATP-dependent multidrug transporters, belonging to the extensive superfamily of ATP-binding cassette (ABC) proteins, holds great significance in MDR [237]. Among these, P-glycoprotein (P-gp), also known as MDR1, stands out as the most widely recognized ABC transporter associated with clinical MDR [237]. These transporters play a pivotal role by preventing the accumulation of toxic concentrations of chemotherapeutic drugs within cancer cells. A promising and innovative strategy to combat drug resistance involves the combination of cytotoxic drugs with non-toxic inhibitors of ABC transporters, often referred to as chemosensitizers [238]. This novel approach opens new possibilities to overcome drug resistance, potentially enhancing the effectiveness of cancer treatment. 

In this context, certain carotenoids serve as substrates for ABC transporters. For example, β-carotene inhibits human P-gp function, enhancing its basal ATPase activity and increasing the sensitivity of the MDR human cervical cancer cell line KB-vin and the MDR human non-small cell lung carcinoma cell line NCI-H460/MX20 to traditional chemotherapeutic drugs [paclitaxel, doxorubicin, etoposide, 5-FU and mitoxantrone] compared to the parental cell lines (HeLaS3 for KB-vin and NCI-H460 for NCI-H460/MX20) [239].

In the case of lutein, it increases the rhodamine-123 (a functional reporter for P-gp) accumulation in the MDR cancer cells Colo-320-MDR (colon cancer), L1210 (murine leukemia), HTB26 (breast cancer) and S180 (murine sarcoma) by inhibiting the efflux pump [154,240,241,242]. This effect also occurs with capsanthin and lycopene treatments in L1210 and HTB26 cell lines [154,240]. Furthermore, in combination with doxorubicin, it synergistically inhibits cell proliferation and tumour growth on the sarcoma cell line S180 [242].

Fucoxanthin mitigates rifampin-induced CYP3A4 and P-gp expression by suppressing pregnane X receptor-mediated CYP3A4 promoter activation and PXR-co-activator interaction in HepG2 cells [243]. The CYP superfamily comprises enzymes that play a role in metabolizing both substances originating outside the body and those produced within it. Nonetheless, the existence of CYP enzymes within cancer cells may exert detrimental effects on chemotherapy medications by facilitating their degradation [244]. On the other hand, fucoxanthin increases rhodamine-123 retention while showing low cytotoxicity and synergistically increases the cytotoxicity of several agents by inhibiting ABC transporters and reducing P-gp expression in resistant Caco-2 and CEM/ADR5000 cells compared to the sensitive parent cell line CCRF-CEM [245]. Moreover, the combined treatment of fucoxanthin and doxorubicin decreases the levels of ABCC1, ABCG2 and ABCB1, while also lowering the activity of CYP3A4, GST and PXR, in multiple drug-resistant cancer cell lines, including breast (MCF-7/ADR), hepatic (HepG-2/ADR) and ovarian (SKOV-3/ADR) cells [246].

Crocin, for instance, enhances the short-term sensitivity of EPG85-257RDB cells to doxorubicin, implying its potential as an adjuvant in cancer chemotherapy [247]. In the case of crocetin, it down-regulates multidrug resistance protein 2, which plays a crucial role in MDR, in human ovarian cisplatin-resistant carcinoma cells A2780-RCIS [248,249].

## 5. Clinical Trials

Carotenoids have been tested in various clinical trials to assess their potential in preventing and treating different health conditions. In cancer trials, the results have sometimes been contradictory when comparing carotenoids and different types of cancer.

In a study with 538 colorectal cancer cases and 564 controls, higher serum levels of α-carotene, β-cryptoxanthin and lycopene were associated with a significantly lower colorectal cancer risk in Guangdong’s Chinese population [250]. However, no significant association was found for β-carotene, lutein or zeaxanthin.

A long-term study indicated that daily multivitamin supplementation with β-carotene, vitamin E and vitamin C resulted in a slight reduction in overall cancer risk, particularly notable among men with a prior history of cancer [251]. Nevertheless, there was no significant impact observed on prostate or colorectal cancer incidence, nor on cancer-related mortality.

The SU.VI.MAX study, involving over 13,000 French adults, found low-dose supplementation with β-carotene reduced total cancer incidence and all-cause mortality in men [252]. This effect was not observed in women, possibly due to differences in baseline antioxidant levels.

A meta-analysis of 19,450 breast cancer cases showed that increased β-carotene intake was associated with improved breast cancer survival [253]. Conversely, other carotenoids, such as lycopene and lutein, did not show notable benefits.

Despite numerous findings, a primary prevention trial with 29,133 male smokers in Finland revealed that daily supplementation with alpha-tocopherol, β-carotene, or both did not decrease lung cancer incidence [254]. Unexpectedly, those receiving β-carotene had an 18% higher lung cancer incidence than those who did not, while α-tocopherol did not reduce total mortality, though more deaths from haemorrhagic stroke occurred in the supplemented group. These findings suggest supplements may have both beneficial and harmful effects, highlighting their complex impact on health, especially for smokers. Additionally, the treatment had no effect on liver cancer or chronic liver disease mortality over 24 years, indicating no risk reduction [255]. Furthermore, β-carotene supplementation altered 17 metabolites, mostly of xenobiotic origin, indicating potential interactions with CYP enzymes, which might explain increased mortality with this supplementation [256].

When using carotenoids as therapeutic agents, it is essential to recognize that, despite their well-known antioxidant properties, they can exhibit cytotoxicity at high concentrations. This cytotoxicity arises mainly from their pro-oxidant activity, which takes place when they are present in excessive amounts, leading to increased oxidative stress and cellular damage [55]. Therefore, understanding these mechanisms is crucial for evaluating the safety and efficacy of carotenoids, especially in high-dose supplementation or specific health contexts.

## 6. Conclusions and Future Directions

Carotenoids, a class of natural pigments commonly found in fruits and vegetables (among other natural sources like micro-organisms, fungi and algae), have demonstrated multifaceted potential as therapeutic agents in the area of cancer. Throughout this review, we have explored their diverse mechanisms of action, shedding light on their role in preventing cancer initiation and progression. Notably, extensive research has been conducted into their ability to induce apoptosis, a well-recognized process of programmed cell death, by modulating factors associated with apoptosis regulation, such as Bcl-2 family proteins and caspases. However, the influence of carotenoids extends beyond apoptosis induction. These compounds have been implicated in affecting several other crucial processes involved in cancer development. Among these, the carotenoids have been linked to alterations in cell cycle progression, effectively halting cancer cell growth. Carotenoids have also displayed antimetastatic properties by interfering with crucial aspects of metastasis, such as cell migration, invasion and the formation of new blood vessels to support tumour expansion. This impact on metastasis, although less explored than the induction of apoptosis, is of great importance, given that metastasis is the leading cause of cancer-related mortality.

Carotenoids have also emerged as potential agents in differentiation therapy, which offers a novel approach to cancer treatment by promoting the maturation of cancer cells into less aggressive phenotypes. Through their involvement in intercellular communication and the regulation of gap junctions, carotenoids can effectively influence the behaviour of cancer cells, potentially mitigating their resistance to therapeutic agents. Furthermore, it is worth noting that carotenoids can interact with mechanisms of drug resistance, a significant challenge in the field of cancer treatment. These compounds have demonstrated their ability to modulate ATP-dependent multidrug transporters, which often play a pivotal role in the resistance of cancer cells to chemotherapeutic agents. Consequently, the use of carotenoids, either alone or in combination with chemosensitizers, has presented a novel strategy to overcome drug resistance and enhance the efficacy of traditional cancer treatments.

All these antitumour phenotypes appear to be due to common mechanisms (Figure 4). Although carotenoids are antioxidant molecules that decrease ROS concentration in normal cells, they exhibit antitumoural mechanisms where they act as pro-oxidants, generating ROS in the tumour context. When carotenoids enter the tumour cell cytoplasm, they increase the concentration of ROS, surpassing the threshold level that cancer cells can tolerate. This surge in ROS production triggers a cascade of events within the cell.

First, this causes damage to cell structures such as DNA, activating death programs and inactivating survival programs. This process involves molecular pathways critical to cellular processes, such as the PI3K/AKT, JAK/STAT, JNK, PPARγ, NF-κB and/or MAPK/ERK pathways. For example, according to the reviewed bibliography, PI3K/AKT and NF-κB are down-regulated after carotenoid treatment, thus inhibiting survival responses. Furthermore, this process activates p53 and p27, which orchestrate the antiproliferative effect, induce apoptosis and inhibit metastasis and angiogenesis.

Regarding the antiproliferative effect, p53 activates p21 and GADD45α, which, along with p27, inhibit the activation of cyclin-CDK complexes, causing cell cycle arrest and producing the antiproliferative phenotype. On the other hand, p53 is a major mediator of apoptosis, triggering the intrinsic apoptosis pathway. This involves the up-regulation of Bax and the down-regulation of Bcl-2 proteins, leading to a decrease in mitochondrial membrane potential and the release of cytochrome c into the cytoplasm, which activates the caspase cascade, culminating in the activation of caspase-3. Additionally, the inhibition of metastatic processes seems to be primarily due to the inactivation of MMP-2 and MMP-9 via p53 and the down-regulation of pathways such as NF-κB.

This mechanism of action shares similarities with doxorubicin. As an anthracycline chemotherapy agent, doxorubicin operates through two primary mechanisms: (i) by intercalating into DNA and disrupting topoisomerase II, which impairs DNA repair, and (ii) by generating free radicals that damage cellular membranes, DNA and proteins [257]. The drug is oxidized into an unstable semiquinone metabolite, which subsequently regenerates doxorubicin while releasing reactive ROS. These ROS effects resemble those produced by the pro-oxidant activity of carotenoids. Notably, carotenoids can enhance the efficacy of doxorubicin by modulating oxidative stress and influencing cellular responses to the drug [258]. While carotenoids generally provide health benefits in healthy tissue, doxorubicin is used as a chemotherapeutic agent to induce cancer cell death, albeit with potential effects on healthy cells.

The fact that carotenoids play a pro-oxidant role in tumour cells and an antioxidant role in healthy cells supports their potential as antitumour agents. This selectivity allows carotenoids to exert cytotoxic effects on tumour cells without affecting healthy cells. Moreover, the dual role of carotenoids is interesting not only from a mechanistic perspective but also for its therapeutic promise. It presents opportunities for combination therapies that leverage these pathways to improve cancer treatment strategies. For example, when combined with chemotherapy, the administration of carotenoids could protect healthy tissues from the side effects of chemotherapy, while enhancing the elimination of cancer cells through a synergistic effect. 

However, some studies have described a decrease in intracellular ROS levels in tumour cell lines treated with carotenoids. For instance, the AGS stomach cancer cell line infected with *H. pylori* exhibited decreased ROS levels after treatment with β-carotene at doses of 0.5 and 1 μM [178]. In contrast, when the same AGS cells, without *H. pylori* infection, were treated with a higher dose of 100 μM β-carotene, there was an increase in ROS, leading to the described antitumour effects [259]. This suggests that the decrease in ROS observed after β-carotene treatment at lower doses could be due to either the dose being too low or the influence of *H. pylori* infection, which modulates ROS production. Similarly, the breast cancer cell line SKBR3 showed reduced ROS levels after treatment with astaxanthin [92]. In the case of SKBR3 cells, the authors noted an imbalance in the enzymes SOD1 and SOD2 due to a reduction in cytosolic ROS, leading to mitochondrial stress and consequent antitumour effects. Finally, in MCF-7 and MDA-MB-231 cell lines treated with lutein at doses of 1–20 μM for 3 h, a decrease in ROS levels was observed [133]. However, this effect appears to be related to the duration of treatment rather than being specific to this type of cancer or carotenoid. In the same cell lines, an increase in ROS levels was reported after treatment with 2 μM lutein, but this time the treatment lasted 24 h [87]. Because all of this, to fully understand the therapeutic potential of carotenoids, it is crucial to describe the circumstances under which their pro-oxidant action occurs and to investigate the ROS homeostasis of tumour cells.

In terms of cancer prevention, inflammation is increasingly recognized as a critical factor in cancer development and progression. Chronic inflammation can create a microenvironment favourable for tumour growth, cause genetic mutations and promote malignant transformation. In this context, carotenoids are known to play an important anti-inflammatory role, including preventing the activation of the pyroptosis pathway and potentially mitigating the deleterious effects of chronic inflammation [260,261].

The recent completion of clinical trials evaluating carotenoid-based interventions in the prevention and treatment of cancer provides valuable insight into this issue. While some trials have demonstrated potential benefits, such as the reduction in cancer incidence with β-carotene supplementation [251,252,253], others have raised concerns, such as the unexpected increase in lung cancer incidence associated with β-carotene supplementation [254]. The relationship between carotenoid intake and health outcomes is multifaceted, influenced by factors like the dose, duration, baseline nutritional status and individual health conditions. These trials emphasize the importance of carefully evaluating the potential risks and benefits of carotenoid supplementation and they underscore the need for continued research to better understand the complex interplay between carotenoids and human health. 

Looking ahead, carotenoid research in cancer involves exploring new combinations. Combinatorial approaches, in which carotenoids are strategically combined with other chemotherapeutic agents, may produce synergistic effects. These combinations may increase the efficacy of conventional treatments, overcome resistance mechanisms and broaden the spectrum of treatable cancers. As we uncover the intricate interaction of carotenoids with various anticancer agents, this review opens the door to new and more potent therapeutic strategies. Furthermore, personalization is another approach on the horizon for carotenoid-based cancer interventions. Since individual factors, such as genetics, metabolism and the particular type of cancer greatly influence treatment outcomes, it would be highly advantageous to customize carotenoid therapies to each patient.

Additionally, innovative drug delivery systems are poised to revolutionize the administration of carotenoids. Among these advanced systems, nanoparticles, liposomes and targeted carriers stand out. Nanoparticles, with their small size and high surface area, can encapsulate carotenoids and facilitate their delivery to specific cellular locations, improving absorption and stability [262]. Liposomes, which are lipid-based vesicles, provide a biocompatible environment that protects carotenoids from degradation and enhances their bioavailability [263]. Targeted carriers, such as conjugated antibodies or ligands, can direct carotenoids precisely to cancer cells, thus maximizing therapeutic effects while reducing exposure to healthy tissues [264]. These advancements address the challenge of carotenoid bioavailability and ensure that these compounds can be delivered efficiently to their intended targets, thereby enhancing their therapeutic impact and minimizing potential side effects. As a result, these systems hold promise for more effective and safer carotenoid-based treatments.

In the field of carotenoid research, increasing attention is being focused on the exploration of new carotenoids, especially those considered rare or less studied like C_5_o carotenoids mainly synthesized by halophilic micro-organisms. While the better-known carotenoids, such as β-carotene, lutein and zeaxanthin have received much attention, emerging compounds, often from unique plants or micro-organisms, offer unexploited potential. Researchers are delving into the depths of biodiversity and discovering carotenoids with distinct structures and functions. These new carotenoids may possess previously unknown anticancer properties, making them valuable candidates for future research.

This is the case of bacterioruberin (2,2′-bis[3-hydroxy-3-methylbutyl]-3,4,3′,4′-tetradehydro-1,2,1′,2′-tetrahydro-γ,γ-carotene-1,1′-diol), a natural carotenoid relatively unknown in the field of cancer research, that is synthesized by halophilic haloarchaea [265]. Organisms such as haloarchaea demonstrate extraordinary resilience under stress. They can withstand intense light exposure, gamma irradiation, DNA damage caused by radiography, ultraviolet irradiation and even hydrogen peroxide [266]. These extraordinary attributes are due to the existence of molecules such as bacterioruberin, which plays a fundamental role as a biological antioxidant, protecting cells against the dangers of oxidative damage. Specifically, bacterioruberin is distinguished by its thirteen pairs of conjugated double bonds, which exceed the nine pairs of β-carotene (Figure 5). This structural superiority of bacterioruberin compared to the most abundant C_40_ carotenoids, gives bacterioruberin an exceptional ability to quench free radicals, placing it above the antioxidant potential of β-carotene [12,25,26]. Thus, bacterioruberin holds intriguing potential for further investigation in the field of cancer research.

Recent research has investigated the cytotoxic effects of bacterioruberin-rich carotenoid extracts from the haloarchaea *Haloarcula* sp. and *Haloferax mediterranei* [267,268]. Specifically, in the case of *H. mediterranei*, known for being the first organism in which CRISPR-Cas was described, cytotoxic effects have been observed on various breast cancer cell lines, including the aggressive triple-negative subtype [267]. This molecule demonstrated selective cytotoxicity toward breast cancer cells while sparing healthy mammary epithelium cells. The study found a significant reduction in the population of breast cancer cells, especially triple-negative ones, and a decrease in cell size following treatment with bacterioruberin-rich carotenoid extracts. These findings provide valuable insights into the potential applications of haloarchaeal carotenoids and advance our understanding of their in vitro effects. Given their potent biological activities, bacterioruberin and related carotenoids are promising natural compounds that could be utilized in developing new strategies and pharmaceutical formulations for combating cancer, enhancing immune response, maintaining skin health and other therapeutic applications. Our group is currently working with bacterioruberin-rich carotenoid extracts from *H. mediterranei* to further explore these potential therapeutic benefits of haloarchaeal carotenoids.

## Figures and Tables

**Figure 1 antioxidants-13-01060-f001:**
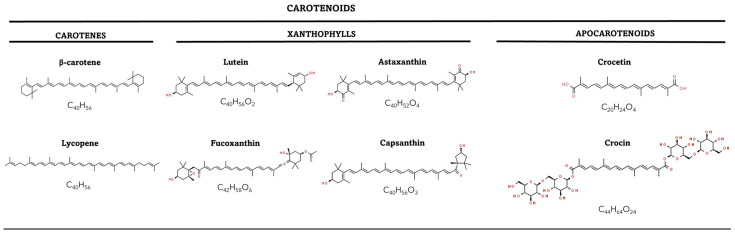
Structure and classification of the most studied carotenoids in cancer. Carotenoids are divided into two main groups, xanthophylls and carotenes. These can also undergo modifications and generate new carotenoids called apocarotenoids. Hydroxyl groups are represented in red. Chemical structures obtained from ChemSpider (Royal Society of Chemistry).

**Figure 2 antioxidants-13-01060-f002:**
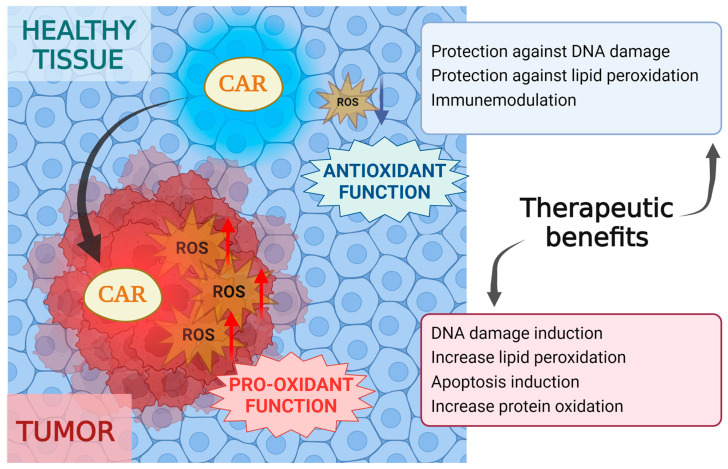
Scheme of the pro-oxidant carotenoid therapy against cancer. Carotenoids exert pro-oxidant effects to selectively target cancer cells, promoting oxidative stress-induced cell death while potentially protecting normal cells. CAR, carotenoid; ROS, reactive oxygen species. Created with BioRender.com.

**Figure 3 antioxidants-13-01060-f003:**
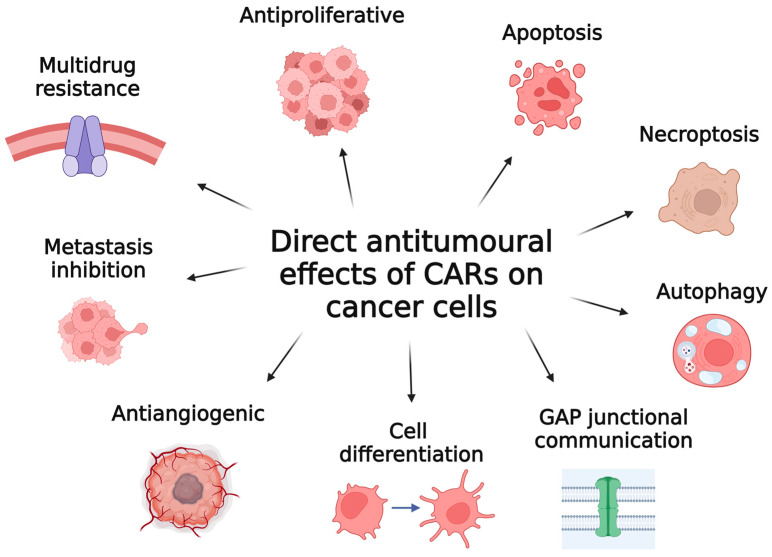
Scheme of direct antitumoural effects of carotenoids on cancer cells. Carotenoids can induce apoptosis, necroptosis, autophagy or cell differentiation; enhance gap junctional communication; and also exhibit antiangiogenic, antimetastatic, multidrug resistance and antiproliferative effects. CAR, carotenoids. Created with BioRender.com.

**Figure 4 antioxidants-13-01060-f004:**
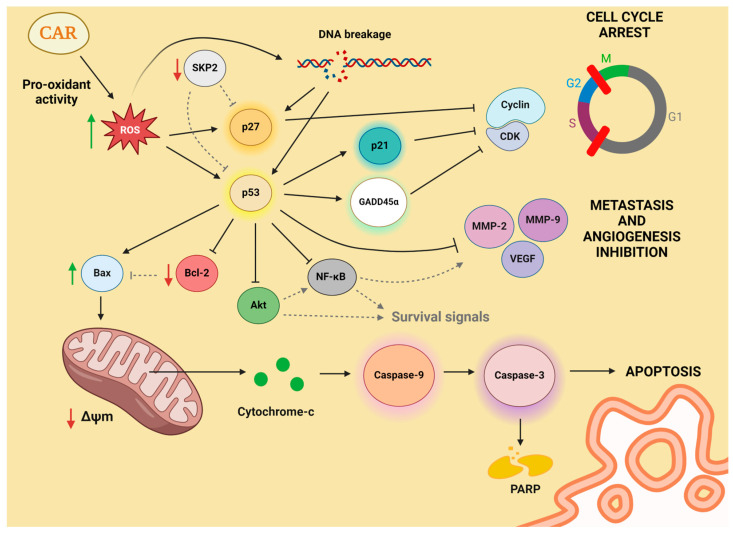
Mechanism of action of the anticancer properties of carotenoids. Graphical representation of the most common mechanisms of action of carotenoids in tumour cells based on the reviewed literature. Arrows with pointed ends indicate activation, while T-bar arrows indicate inhibition. ΔΨm, membrane potential; Akt, protein kinase B; Bax, Bcl-2-associated X-protein; Bcl-2, B-cell lymphoma 2 protein; CAR, carotenoid; CDK, cyclin-dependent kinases; GADD45α, growth arrest and DNA-damage-inducible protein 45 α; MMP, matrix metalloproteinase; NF-κB, nuclear factor-kappa B; PARP, poly ADP-ribose polymerase; ROS, reactive oxygen species; SKP2, S-phase kinase-associated protein 2. Created with BioRender.com.

**Figure 5 antioxidants-13-01060-f005:**
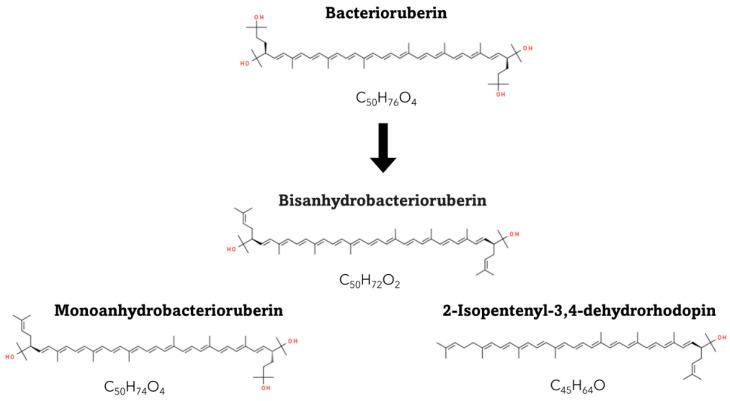
Chemical structure and formulation of bacterioruberin and its derivatives. Bacterioruberin can be modified and converted into bisanhydrobacterioruberin, monoanhydrobacterioruberin or 2-isopentenyl-3,4-dehydrorhodopin. Hydroxyl groups are represented in red. Chemical structures obtained from ChemSpider (Royal Society of Chemistry).

## Data Availability

No new data were created or analyzed in this study. Data sharing is not applicable to this article.

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
