# Peer review of "The Antitumour Mechanisms of Carotenoids: A Comprehensive Review"

_antioxidants, 2024, doi:10.3390/antiox13091060_

Round 1

Reviewer 1 Report

This manuscript reviews anti-tumor mechanisms of carotenoids. The authors explore their role in inducing apoptosis, inhibiting cell cycle progression and preventing metastasis by affecting oncogenic and tumor suppressor proteins. Some carotenoids can exhibit pro-oxidant effects under certain conditions and would be able to elevate reactive oxygen species levels in tumoral cells. Clinical trials highlight the conflicting results of carotenoids in cancer treatment and the importance of personalized approaches. The review is interesting. 

1.    The carotenoids can be used to reduce and enhance the ROS, which is very intriguing, the authors should discuss this problem.

2.    The cytotoxicity of carotenoids should be added in the manuscript.

3.    The carotenoids carried by drug delivery system should be discussed.

4.    The targeting effect of carotenoids should be discussed.

5.    The main mechanism of cytotoxicity of should be compared with other chemotherapeutic agents. The solubility of carotenoids should be discussed.

6.    The authors should cite more recent references.

7.    Please carefully check the text for writing and grammar errors.

Reviewer 2 Report

The present review is interesting and presents valuable and actual information so readers can use it in the area where it may be cited.  However, some references are missed, for instance, doi: 10.1080/17518253.2021.1998648 

Not applicable

Reviewer 3 Report

This review provides a comprehensive summary of the top-studied carotenoids in terms of their biochemical properties and multifaceted anticancer mechanisms. The manuscript is excellently written, with well-organized citation styles and high-quality figures and tables. The reviewer highly commends the authors for their dedication in collecting and composing this extensive review, which not only summarizes the current progress in carotenoids for cancer treatment but also offers valuable insights into future directions for carotenoid applications in cancer and other diseases.

1. Line 85: The inconsistent use of brackets and parentheses appears confusing.

2. Lines 90&155: There is no need to abbreviate 'carotenoid cleavage dioxygenases' or 'β-carotene 15,15'-dioxygenase' since these terms appear only once in the manuscript. In academic writing, generally, if a term appears more than once, write out the full term followed by the abbreviation in parentheses the first time it is mentioned. Use only the abbreviation for all subsequent mentions. Please conduct thorough proofreading to eliminate any redundancy in this regard, as the review is quite long and excessive use of abbreviations might interfere with readability.

3. Lines 112-114: Please specify the full term of “CAR” above the formula to clarify this abbreviation.

4. Lines 133-161: The authors discussed the bioavailability of carotenoids. Could the authors clarify whether the general bioavailability of carotenoids is considered high or low? How does first-pass metabolism impact their concentration in plasma? Are there any studies specifically investigating this aspect?

5. Lines 171-177: This statement is overly general as some cancers can upregulate oxidative phosphorylation instead of enhancing glycolysis. So, the metabolic preference can vary depending on the type of cancer, its stage, and the tumor microenvironment. Given that mitochondria are the primary source of ROS production, please consider expanding the discussion regarding this variability.

6. Line 241: The figure legend uses 'antimultidrug resistance,' while the figure itself uses 'multidrug resistance.' Please revise Figure 3 to ensure consistency and avoid confusion.

7. Lines 258-259: This sentence contains a grammatical error due to a missing “and”.

8. Lines 287-292: This sentence appears lengthy and convoluted. Please consider breaking it down.

9. The authors primarily presented a large number of preclinical studies conducted on cancer cell lines with diverse molecular mechanisms being uncovered. It appears that there is no inclusion of current studies conducted on animal models, revealing a significant gap in the translational potential of carotenoids in cancer treatment. Why did the authors selectively focus on discussing research on the cell culture level? Have there been any animal studies evaluating the efficacy of carotenoids?
